# Graphene-Based Biosensors for Detection of Biomarkers

**DOI:** 10.3390/mi11010060

**Published:** 2020-01-03

**Authors:** Yunlong Bai, Tailin Xu, Xueji Zhang

**Affiliations:** Research Center for Bioengineering and Sensing Technology, School of Chemistry and Biological Engineering, University of Science and Technology Beijing, Beijing 100083, China; b532894237@126.com

**Keywords:** graphene, graphene derivative, biosensor, detection, biomarker

## Abstract

The development of biosensors with high sensitivity and low-detection limits provides a new direction for medical and personal care. Graphene and graphene derivatives have been used to prepare various types of biosensors due to their excellent sensing performance (e.g., high specific surface area, extraordinary electronic properties, electron transport capabilities and ultrahigh flexibility). This perspective review focuses on graphene-based biosensors for quantitative detection of cancer-related biomarkers such as DNA, miRNA, small molecules and proteins by integrating with different signal outputting approaches including fluorescent, electrochemistry, surface plasmon resonance, surface enhanced Raman scattering, etc. The article also discussed their challenges and potential solutions along with future prospects.

## 1. Introduction

According to the World Health Organization, there will be 24 million new cancer cases and 14.5 million cancer-related deaths each year by 2035 [1]. Early diagnosis is extremely vital to improve the survival rate of cancer patients [2,3,4]. The incidence of cancer is closely related to the content of specific biomarkers, which are usually found in body fluids (such as blood or urine), or in cells and tissues. Quantitative detection of biomarkers plays a crucial role in early screening, clinical testing, and evaluation of therapeutic effects [5,6,7]. Biosensors that combine a sensitive biological component including enzymes, antibodies, nucleic acids, etc., with a physicochemical detector can be used for the detection of various targets according to the change of signal strength upon the interaction of a biological element with analyte [8,9]. Such biosensors also enable fast and sensitive detection of biomarkers to meet clinical testing and scientific research needs [10,11,12,13].

Recently, graphene has been drawing tremendous attraction owing to the outstanding feature of electrochemical, adsorption performance, mechanical strength and flexibility to serve as an attractive candidate for biosensors [14,15,16,17]. Graphene, formed by carbon atom hybridization with sp^2^ electron orbital, has a high specific surface area, excellent electron transport capabilities and strong mechanical strength, which is essential for constructing biosensors. This review gives a detail of classification and characteristics of graphene, and a detail of the application of various types of graphene and graphene derivatives-based methods with diverse signals outputting approaches to achieve quantitative detection of different kinds of biomarkers including DNA, microRNA, small molecules and proteins. We also comprehensively summarized and compared the detection principle, target molecules, detection limits and detection range of graphene-based different sensors in recent years. Such graphene-based biosensors are expected to show great potential in biological analysis and clinical medicine.

## 2. Characteristics and Classification of Graphene

Graphene-based nanomaterials mainly include graphene, graphene oxide (GO) and reduced graphene oxide (rGO) as demonstrated in Figure 1 [18]. Graphene is a two-dimensional carbon material with a thickness of a single atomic layer. GO is a functionalized graphene obtained by oxidative stripping of graphite, which is roughly similar to the structure of graphene. Compared with graphene, GO contains various oxygen-containing functional groups such as C–O–C, –COON, –OH and C=O, which has relatively strong reactive activity, good dispersibility and favorable binding sites for future functionalization [19,20]. By chemical or heat removing oxygen-containing functional groups of GO, the obtained rGO also has the characteristics of GO, such as good thermal conductivity, good chemical stability, excellent mechanical properties, high electron mobility and a large specific surface area.

Graphene has been making a tremendous influences in several research areas due to its unique physical and chemical properties [21,22]. There are several advantages of such graphene-based sensors for sensing, including the following:

(1)High specific surface area. 2630 m^2^/g for single-layer graphene theoretically, which gives rise to high densities of attached recognition component or analyte molecules. It contributes to high detection sensitivity and the miniaturization of the device.(2)Excellent electronic properties and electron transport capabilities. The carbon atoms of graphene hybridized in the form of sp2 constitute a huge π–π conjugate system in which the electrons are freely moving. These properties make graphene a candidate in the field of electrochemical sensing.(3)Strong mechanical strength and pliability. Single-layer graphene possesses a thickness of ~0.335 nm, the hardness of which is higher than diamond due to strong C=C bonding in the atom plane; while opposite to diamond, the interlayer bonding via Van der Waals forces makes it a soft material. This will greatly benefit the development of wearable sensor devices.

## 3. Applications of Different Types of Graphene-Based Biosensors

Graphene and graphene derivatives with properties of a large specific surface area, high electron transport rate and high temperature resistance can be used as a signaling device or carrier of biometric components to achieve a quantitative detection of biomolecules, which is described in detail in the following sections. However, there are more than 10,000 publications about five types of graphene-based sensors (Fluorescence, FA (fluorescence anisotropy), Electrochemistry, SPR (surface plasmon resonance), SERS (surface enhanced Raman scattering)). In this manuscript, we only focused on the recent graphene-based biosensors. Table 1 summarized the graphene-based biosensors for various targets by integrating with signal output types including fluorescent, electrochemistry, SPR and SERS.

### 3.1. Graphene-Based Fluorescent Biosensors

Fluorescence is the emission of light by a fluorescent tag with labeled targets that have absorbed external incident light, which is a commonly used detection technique in biological monitoring owing to the high sensitivity, low detection limit, good accuracy, etc. [47]. Graphene and its derivatives have high surface-to-volume ratio and highly distance-dependent fluorescence quenching ability based on fluorescence resonance energy transfer (FRET), making them universal carriers and quenchers for fluorescence recognition probes or targets.

GO-based fluorescent sensors were firstly introduced for the sensitive and selective detection of DNA and proteins by Lu et al. The presence of target ssDNA or protein lead to the binding of dye-labeled ssDNA and target DNA or aptamer and target protein, releasing the ssDNA and aptamer from GO, thereby restoring the dye fluorescence, which enable the detection of target ssDNA and thrombin [48]. The same biosensor was also reported to detect vascular endothelial growth factor (VEGF) with a detection limit of 0.25 nM [49]. In addition, Chen et al. reported a GO-based fluorescent biosensor for quantitative detection of dopamine in biological matrixes [50]. Such a detection approach was based on self-assembly of dopamine on the surface of GO by multiple noncovalent interactions, and significant fluorescence quenching suggested its ability as a label-free fluorescence biosensor for detection of dopamine with a detection limit of 94 nM. Quan et al. introduced a graphene quantum dots (QDs)-based fluorescence sensor for high selectivity and sensitivity detection of ssDNA as shown in Figure 2 [51]. Unique FRET between the QDs and GO occurred when QDs-labeled ssDNA were loaded into GO. The exist of target DNA would cause the liberation of ssDNA-QDs from GO, and finally resulted in the fluorescence recovery. The detection limit of this fluorescence sensor was 75 pM DNA.

Enzyme amplification technology can be applied to graphene-based fluorescent sensors. As demonstrated in Figure 3, He et al. proposed a simple aptamer-based fluorescent sensor for adenosine triphosphate (ATP) detection by exonuclease III (Exo III) digestion and GO assistance [23]. This method used aptamer as an affinity element to identify ATP, as the amplification capability of Exo III and the adsorption characteristics of graphene contributes to sensitive detection of ATP with a detection limit of 31 nM. Xia et al. reported a GO-based fluorescent aptasensor for simultaneous detection of telomerase and miRNA in living cells and tissue samples [52]. Template-strand primer and fluorophore-labeled telomerase/miRNA oligonucleotides were loaded onto GO. In the presence of targets, the double-stranded oligonucleotides would be away from the GO surface, leading to obvious fluorescence recovery.

Fluorescence anisotropy (FA) assay is an analytical method based on the rotation speed of fluorophore-induced signal differences upon the binding of targets [53]. GO can be used as the signal amplifier to fabricate FA biosensor. The fluorophore exhibited very high anisotropy when bound to GO, owing to the larger volume of GO. Conversely, low anisotropy was observed when the fluorophore was dissociated from GO. Liu et al. demonstrated two kinds of GO-based FA sensors for the detection of ATP [54]. As for the signal-off detection approach (Figure 4A), high anisotropy was observed when aptamer bound to GO, while the FA was greatly reduced due to aptamer combined with ATP, where the decreased FA signal allowed the detection of target. In contrast, a dye-labeled ssDNA was used as a signaling transduction probe, which was partially complementary to the capture aptamer to form a DNA duplex with a relatively low anisotropy signal. The presence of target ATP induced the release of dye-labeled ssDNA, which bound to GO to amplify the mass, thus leading to an increased anisotropy signal as shown in Figure 4B. Xiao et al. reported a novel GO amplified FA sensor for the quantitative detection of a variety of target molecules (ssDNA, adenosine and thrombin) [55]. Dye-labeled-ssDNA was immobilized on GO through a double-stranded region with the capture DNA, and such a design allowed an appropriate distance between GO and fluorescent dye, which restricted the rotation of dye with the entire formation to get a high FA signal and decreased the quenching efficiency of GO towards the dye. In the presence of target DNA, a toehold-mediated strand exchange reaction occurred, and a decreased FA signal was observed, with the detection limit being 4.6 nM DNA. The same analytical methods were also applied to detect adenosine and thrombin with good accuracy and sensitivity.

Enzyme amplification technology was applied in a graphene-based FA biosensor. Huang et al. established two different FA sensors based on nicking enzyme amplification and GO enhancement for sensitive detection of biomolecules including adenosine and thrombin in homogeneous solution [56]. As demonstrated in Figure 5, the binding of aptamer and the target induced the hybridization between the dye-labeled DNA probe linked to GO and aptamer probe, resulting in forming a nicking site-containing duplex DNA region, which further triggered releasing of dye-labeled short DNA from GO. The obviously decreased FA value was used to detect adenosine with a detection limit of 2 pM.

Graphene-based fluorescent biosensors have ultrahigh sensitivity and can achieve the quantitative detection of biomolecules in complex samples. However, the labeling of fluorophores and the reading of signal data by spectrometers are relatively complicated and cumbersome. Graphene-based FA sensors are a potential candidate for rapid and efficient detection methods without interference of background. Whereas, this method requires a module in fluorescence spectrometer for measuring the anisotropic signal, which is more complicated and expensive.

### 3.2. Graphene-Based Electrochemical Biosensors

Electrochemical sensors depend on the change of current, potential or impedance upon the occurrence of reaction on the surface of an electrode. Electrochemical methods have received much attention due to the relative low-cost, simplicity, low detection limits and ease of miniaturization [57]. The large specific surface area and excellent electrical conductivity of graphene and graphene derivatives allow the protein adsorption and rapid electron transfer between the redox centers and the surface of the electrode, which ensure the accurate and selective detection of target biomolecules.

The sensitive detection of DNA and miRNA, which play vital roles in storing and transmitting genetic information, are valuable in bioanalysis. Guo et al. reported a graphene-based electrochemical biosensor for detection of DNA with ultrahigh sensitivity with a detection limit of 9.4 zM [58]. Huan et al. proposed a label-free electrochemical biosensor for rapid detection of DNA based on gold nanoparticles (AuNPs)-toluidine blue-GO nanocomposites with a detection limit of 2.95 pM [59]. A label-free ultrasensitive detection of miRNA electrochemical sensor based on sandwiched silver nanoparticles (AgNPs) in polyaniline and N-doped graphene was also developed [27]. Such a graphene-based biosensor achieved high sensitivity detection of target miRNA with a wide dynamic detection range from 10 fM to10 µM and a low detection limit of 0.2 fM.

A variety of graphene-based immunoelectrochemical biosensors are constantly being developed for sensitive detection of cancer-related proteins. Chih et al. proposed a reusable immunosensor based on a GO-modified Au electrode. The use of GO enabled an effective carrier for the avastin antibody, and a changing of amperometric signal enabled efficient detection of VEGF in human plasma with a detection limit of 31.25 pg/mL [60]. As shown in Figure 6, He et al. fabricated a platform by electrophoretic deposition rGO onto a gold electrode with post-functionalized folic acid as ligand for the sensitive detection of folic acid protein with a detection limit of 1 pM. Upon the target binding, a significant decreasing of current can be measured using differential pulse voltammetry [25]. Elumalai et al. presented a graphene-based amperometric sensor by using a nanoporous graphene-modified electrode as demonstrated in Figure 7. Preparation of nanoporous graphene was done via fast reduction of chemically-prepared GO in an acidic solution using Mg/Zn bimetal strips. Owing to its excellent electrocatalytic activity, GO (Mg/Zn) composes showed enhanced electrochemical response in the addition of folic acid analyte [26]. Castillo et al. described a peptide nanotube-folic acid-modified graphene electrochemical biosensor for detection of human cervical cancer cells. The recognition of human cervical cancer cells was based on the specific binding of folic acid and the folate receptors expressed in cell surface [61]. Erika et al. described a label-free graphene-based electrochemical immunosensor for detection of cystatin C. The detection limit of this immunosensor was 0.03 ng/mL [28]. Assari et al. reported an electrochemical immunosensor for detection of prostate specific antigen (PSA) by the using of rGO and AuNPs decorated glassy carbon electrode [30]. Lv et al. presented an electrochemical sandwich immunosensor by using nitrogen/sulfur co-doped GO and Au@Ag nanocubes [31]. Such a sandwich immunosensor achieved detection of cardiac troponin I (cTnI) with a linear range from 100 fg/mL to 250 ng/mL and a detection limit of 33 fg/mL. Some graphene-based electrochemical sensors do not depend on the antigen-antibody reaction. Figure 8 demonstrated spindle-like palladium nanoparticles (PdNPs) functionalized GO-cellulose microfiber electrochemical sensors for detection of dopamine with a detection limit of 23 nM [34]. A Fe_3_O_4_ and polyaniline-modified GO electrochemical biosensor for non-enzymatic sensing of glucose was reported. The large surface of GO and Fe_3_O_4_ along with the enhanced charge transfer capability of polyaniline provided a pronounced electrochemical response, which enabled the detection of glucose with a detection limit of 10 nM [29].

Simultaneous detection of multiple targets can be performed by modifying diverse recognition molecules on the surface of a graphene-based electrode. A sandwich-format electrochemical immunosensor for simultaneous determination of alpha-fetoprotein (AFP) and carcinoembryonic antigen (CEA) was fabricated by using carboxyl graphene nanosheets as sensing probes, which were functioned by immobilization of toluidine blue and antibody of CEA and AFP [62]. Figure 9 presented a novel GO/ssDNA/poly-l-lactide nanoparticles-based electrochemical biosensor. The ssDNA-functionated GO was firstly decorated in the electrode, the presence of VEGF enabled immobilization of dual-antibody-modified nanoparticles, which further captured more targets as an amplifier. Such a GO-based electrochemical biosensor was used for simultaneously detecting VEGF and PSA in human serum for early diagnosis of prostate cancer [33]. A graphene flowers-modified electrochemical sensor for simultaneous determination of three targets was developed. The graphene flowers-modified electrode exhibited high electrocatalytic activities, which was a great benefit to the simultaneous detection of ascorbic acid, dopamine and uric acid in urine with good selectivity and sensitivity [63].

Graphene-based electrochemical biosensors are the most widely used devices which measure the changed electrical signals caused by the electrons produced by the chemical reactions between the target and biorecognition element. The main advantages of such biosensors are low prices, sensitivity and quickness, and the future direction of graphene-based electrochemical biosensors are high throughput, portable and miniaturized.

### 3.3. Graphene-Based SPR Biosensors

Surface plasmon resonance (SPR) is widely used in biological and chemical investigations to study molecular interactions. Typically, the bio-recognition molecules were modified on the surface of metal (especially gold or silver) film. When the target molecules exist, the surface of the metal film changes upon the binding of targets and recognition molecules. SPR enables the quantitative detection of target molecules and offers special advantages including label-free for test, in-situ detection, ease of fabrications, low cost and short analysis time compared with other analytical methods [64]. Graphene or its derivatives with a large specific surface area can carry more biometric molecules including antibody and functional DNA, which have been generally used for enhancing SPR signals compared with bare metal thin films [65,66].

The first application of graphene-Au-based SPR sensor was depending on the interaction between thrombin and the specific aptamer-modified rGO surface, where the changing of SPR signal enabled the sensitive detection of thrombin with the detection limit of 50 pM [67]. A SPR-based sensing of lysozyme in serum through a bacteria-modified GO surface was reported as shown in Figure 10 [35]. Upon addition of lysozyme, the integrity of the bacterial cell wall was affected, together with a significant change in morphology on the GO-based interfaces, causing a characteristic decrease in the SPR signal. DNA and RNA can be modified into graphene-based metal SPR membrane surfaces for biomarker sensing. A graphene/Cu SPR surface for the detection of ssDNA was reported. The sensor contained a graphene-coated SPR interface and ssDNA-functionalized gold nanostars. When the target DNA existed in the solution, interaction with ssDNA occurred. The desorption of the nanostructures from the graphene matrix produced a changing SPR signal, which allowed the sensitive detection of DNA with a detection limit of 500 aM [68]. Xue et al. reported a SPR platform for direct sensing of DNA/GO binding (Figure 11). The AuNPs-ssDNA was designed as a competitive reagent to amplify the SPR signal. This sensor could be used for sensitive detection of ssDNA with a detection limit of 10 fM [69]. A high sensitivity SPR biosensor was constructed based on GO-AuNPs composites. The dual amplification strategy based on DNA-GO-AuNPs- functionalized sensor chip and upper layer to perform the sensitive detection of miRNA and small molecule adenosine with the detection limit of 0.1 fM for miRNA and detection limit of 0.1 pM for adenosine [36].

Multiple graphene-based SPR sensors depend on the interaction between antigen and antibody and were developed. For example, a GO-silver-coated polymer cladding silica fiber optical SPR biosensor for detection of human immunoglobulin G (IgG) was presented. The existence of GO enhanced the intensity of a confined electric field surrounding the sensing layer, which enabled the detection of IgG with a detection limit of 40 ng/mL [37]. Figure 12 presented a highly sensitive GO-based SPR immunosensor for the detection of cytokeratin 19, a biomarker of lung cancer. The functional group on the surface of the GO could fix a lot of antibodies. The presence of targets enabled obvious SPR changing, which allowed sensitive detection of cytokeratin 19 with a detection limit of 1 fg/mL [40]. Pang Ng et al. presented a nickel-doped graphene localized SPR biosensor for detection of 3-nitro-l-tyrosine (3-NT), a biomarker of neurodegenerative diseases. As the Ni-doped graphene favors chemical adsorption of 3-NT molecules by forming the metal-nitro bond, capture of 3-NT by direct chemisorption was used in this assay [39].

### 3.4. Graphene-Based SERS Biosensors

Surface enhanced Raman scattering (SERS) is a surface-sensitive technique that enhances Raman scattering by molecules adsorbed on rough metal surfaces. SERS is an indispensable technique to detect the molecules with high sensitivity and selectivity [70]. Graphene, having an ideal flat surface and strong chemical interaction with plenty of target molecules, can be used as Raman enhancement substrates. Graphene-based SERS sensors were demonstrated to be promising for quantitative and repeatable detection of trace target biomolecules [71,72].

A graphene-based SERS immunoassay was usually applied to the quantitative detection of different kinds of proteins. For example, Ali et al. reported a nanographene-mediated metallic nanoparticle clusters for SERS biosensing of IgG with a detection limit of 31 fM [41]. Yang et al. developed a GO and AgNPs-based platform for sensitive SERS immunoassay of PSA. The enzyme label of this method controls the dissolution of AgNPs on the surface of GO through hydrogen peroxide, which was produced by the oxidation of the enzyme substrate. With the dissolution of AgNPs, a greatly decreased SERS signal of GO was obtained [42]. A GO-AuNPs-based amplification SERS immunoassay for detection of cTnI was demonstrated in Figure 13 [44]. Antibody and Raman reporter-labeled AuNPs-functionalized GO were employed as both SERS nanotags and signal amplification carriers. Antibody-modified magnetic beads were applied as the capture probe and separation agents. In the presence of cTnI, sandwich-type immunocomplexes formed through antibody-antigen interactions. Due to the strong SERS enhancement ability of the designed GO/AuNP complexes and a high binding chance between cTnI and the GO/AuNP complexes, the proposed SERS-based immunoassay achieved the detection of cTnI with a detection limit of 5 pg/mL. Demeritte et al. reported a hybrid GO-based plasmonic-magnetic multifunctional SERS immune sensor for selective separation and label-free identification of Alzheimer’s disease biomarkers (*β*-amyloid) [73]. Zhou et al. developed a sensitive and highly selective GO-based aptasensor for SERS detection of ATP [43]. Two split aptamers of ATP were used as specific recognition elements. The first one was attached to the AuNPs-decorated GO, and the other one was placed on AuNPs. The introducing of ATP can form a sandwich structure that brought the GO/Au nanolayer and the AuNPs in close proximity, and the enhanced SERS signal allowed the sensitive quantification of ATP with a detection limit of 0.85 pM.

GO and AuNPs-based SERS platforms for DNA biosensing was reported (Figure 14) [45]. The capture probe 1 was functionalized by GO-AuNPs and capture probe 2 was modified by Raman dye and AuNPs. The presence of target DNA induced the hybridization between capture probe 1, target and capture probe 2. Coupling of the two capture probes generated a locally enhanced electromagnetic field, which provided significant amplification of the SERS signal. Therefore, employing the short-length probe DNA sequences and two SERS active substrates, such a sensor has managed to improve the sensitivity of the biosensors to achieve a lowest limit of detection as low as 10 fM. Furthermore, the biosensor also showed excellent performance to discriminate six closely related non-target DNA sequences, and exhibited sensitivity for single nucleotide base-mismatch in the target DNA. Although the developed SERS biosensor would be an attractive platform for the authentication of, for example, a Malayan box turtle from diverse samples including forensic and/or archaeological specimens, such a sensor shows universal application for detecting gene specific biomarkers for cancer detection.

### 3.5. Graphene-Based Multifunction Detection Sensor

Recently, smart multifunction sensors that output multiple detection signals on one system were reported [74]. Ouyang et al. showed a new dual-spectroscopic all-in-one strategy for quantitative determination of aristolochic acids in complex biological sample matrix. As shown in Figure 15, aristololactam (AAT), a bioproduct of aristolochic acid I (AAI), was directly detected by fluorescence spectrometry after extraction, while AAI was detected by SERS with a graphene-enhanced absorption and magnetic retrieval separation strategy [46]. Our group had built a versatile graphene and nanodendritic gold-based all-in-one biosensor that could perform SERS, with an electrochemical and fluorescence tri-modal miRNA detection in a single microdroplet system (Figure 16) [75]. The existence of graphene led to the changed fluorescence signal output due to the disparate absorption behavior of double and single nucleic acid strands. Meanwhile, benefiting from the large surface area of a nanodendritic gold electrode, sensitive electrochemical and SERS detection of targets could be achieved. The novel all-in-one sensors with simple and fast features had significant implications for use in precise physiological diagnosis and real-time monitoring.

## 4. Conclusions and Outlooks

In conclusion, the recent progresses of graphene and its derivatives-based biosensors are reviewed including the design strategies and detection results. These biosensors always functioned with a DNA probe, antibody, aptamer, protein or small organic molecules for target recognition. Sensitive detection of DNA, miRNA, small molecules, and proteins have been achieved through the integrating of multiple outputting signals (fluorescent, electrochemistry, SPR and SERS).

Even though a large number of graphene-based sensors reported in the literature exhibited good stability and repeatability, performance of some sensors in actual biological samples (such as blood and urine) often failed to achieve the desired detection results, which was mainly due to some non-specificity of biological and chemical molecules during the interfacial reaction of graphene and targets. Although some sample preparations (e.g., separation or preconcentration) are necessary before the final tests for biological samples, however, they always face the limits of complex procedures and are a time-consuming operation [76]. One effective solution to such problems is to develop ultrasensitive and high-specific sensors. Those challenges remain to be overcome by continuous effort to achieve large-scale production of graphene-based sensors. Nowadays, graphene-based all-in-one sensors have received extensive attention, as described in Section 3.5, which greatly expanded the application range of such sensors and provided new ideas for multiple target detection analytical methods. These multifunction sensors make a combination of multiple biomarkers recognition and different signal readouts. The future development direction of graphene-based biosensors should be more portable, reproducible, miniaturized and high-throughput in detection.

## Figures and Tables

**Figure 1 micromachines-11-00060-f001:**
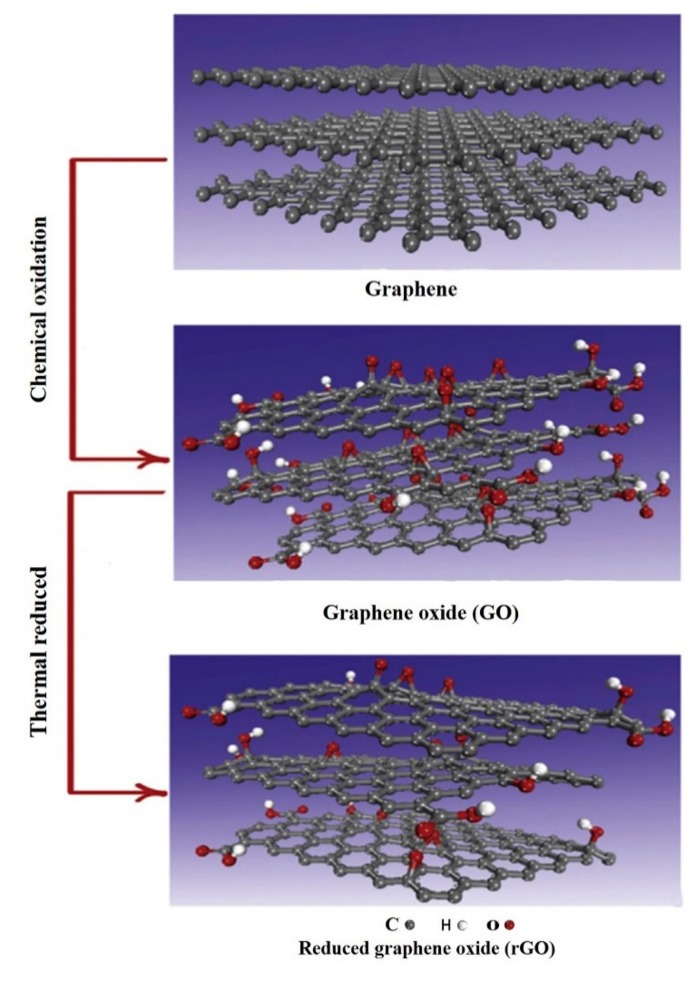
Graphene and graphene derivatives including graphene oxide (GO) and reduced graphene oxide (rGO). Reprinted with permission [18]. Copyright 2019, Elsevier B.V.

**Figure 2 micromachines-11-00060-f002:**
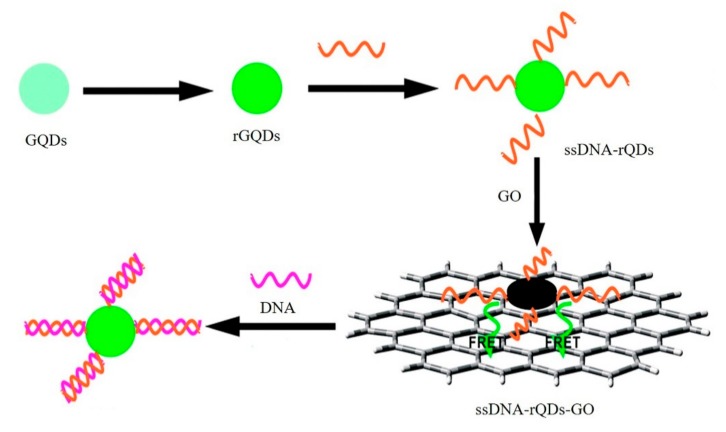
Graphene quantum dots (QDs)-based fluorescence sensor for high selectivity and sensitivity detection of ssDNA. Reprinted with permission [51]. Copyright 2014, Royal Society of Chemistry.

**Figure 3 micromachines-11-00060-f003:**
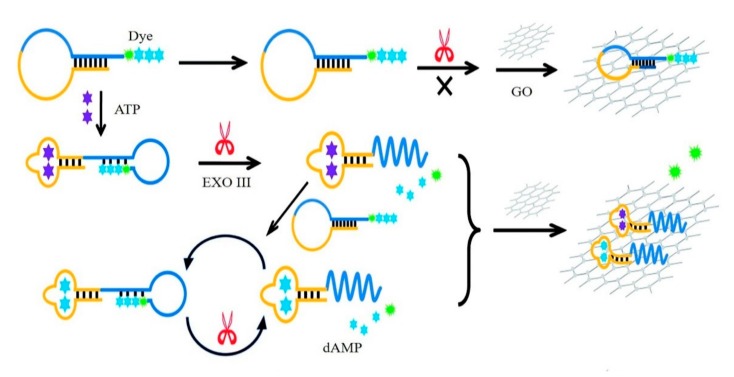
GO-based amplified fluorescent aptasensor for detection of ATP (adenosine triphosphate). Reprinted with permission [23]. Copyright 2017, Royal Society of Chemistry.

**Figure 4 micromachines-11-00060-f004:**
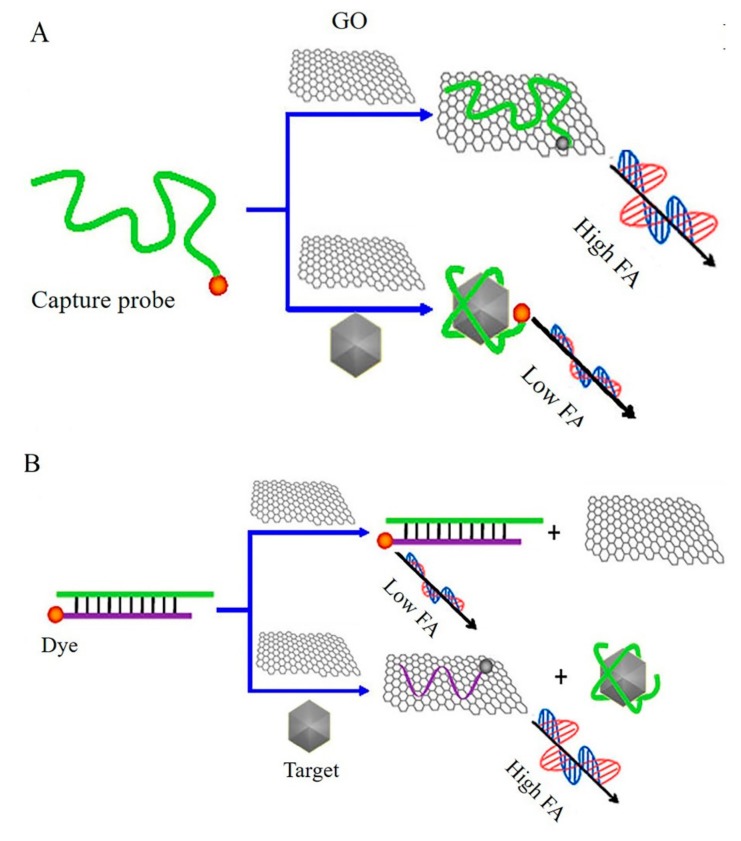
Graphene-based fluorescence anisotropy (FA) biosensors for detection of ATP by (**A**) signal-off signal and (**B**) signal-on signal. Reprinted with permission [54]. Copyright 2013, American Chemical Society.

**Figure 5 micromachines-11-00060-f005:**
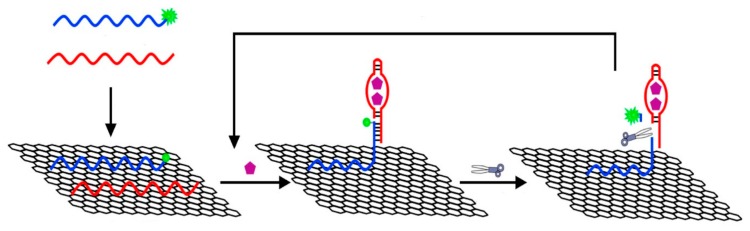
Graphene-based FA biosensors for detection of adenosine. Reprinted with permission [56]. Copyright 2014, Elsevier B.V.

**Figure 6 micromachines-11-00060-f006:**
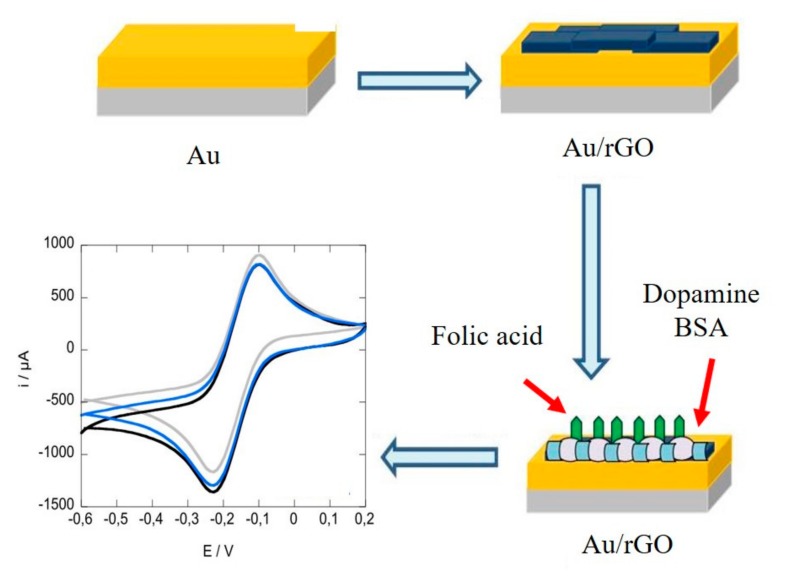
Electrochemical biosensor using graphene-modified electrode for quantitative detection of folic acid protein. Reprinted with permission [25]. Copyright 2016 Elsevier B.V.

**Figure 7 micromachines-11-00060-f007:**
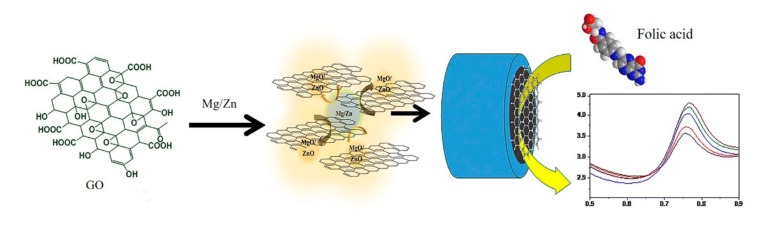
Graphene-based electrochemical biosensor for detection of folic acid. Reprinted with permission [26]. Copyright 2016 Springer.

**Figure 8 micromachines-11-00060-f008:**
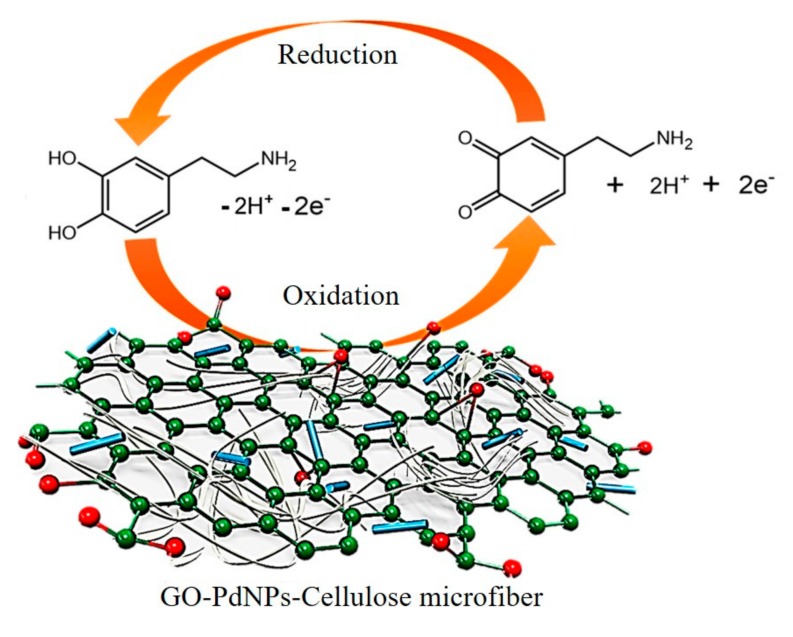
Electrochemical biosensor using graphene-modified electrodes for detection of dopamine. Reprinted with permission [34]. Copyright 2019 Elsevier B.V.

**Figure 9 micromachines-11-00060-f009:**
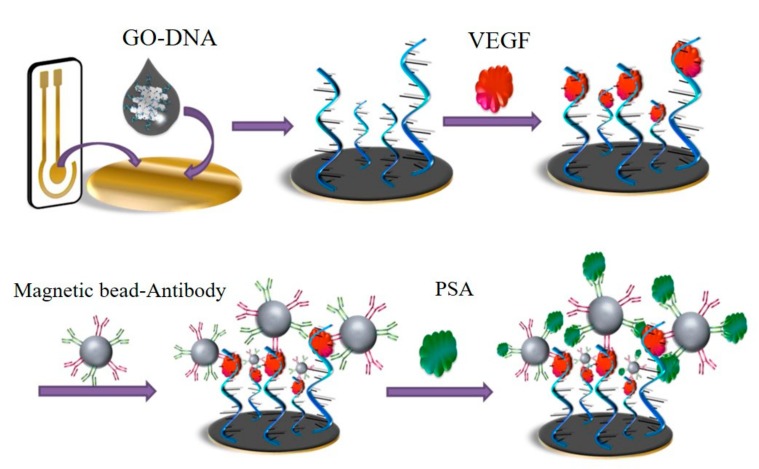
GO-based electrochemical biosensor for simultaneous detection of VEGF and PSA. Reprinted with permission [33]. Copyright 2016 Elsevier B.V.

**Figure 10 micromachines-11-00060-f010:**
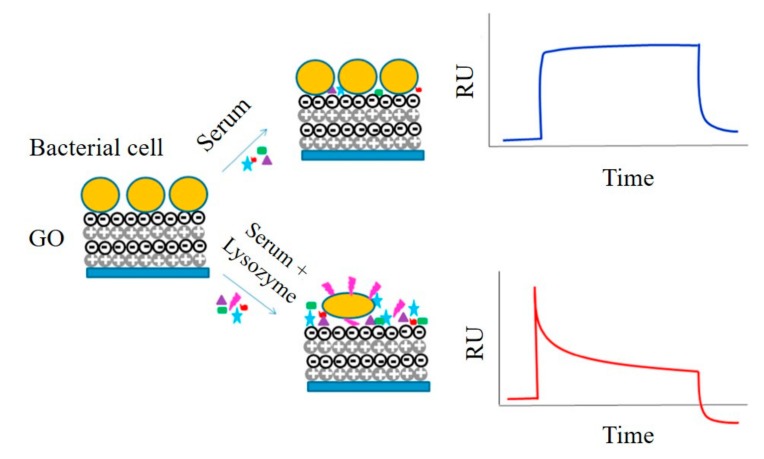
Graphene-based SPR biosensors for quantitative detection of lysozyme. Reprinted with permission [35]. Copyright 2017, Elsevier B.V.

**Figure 11 micromachines-11-00060-f011:**
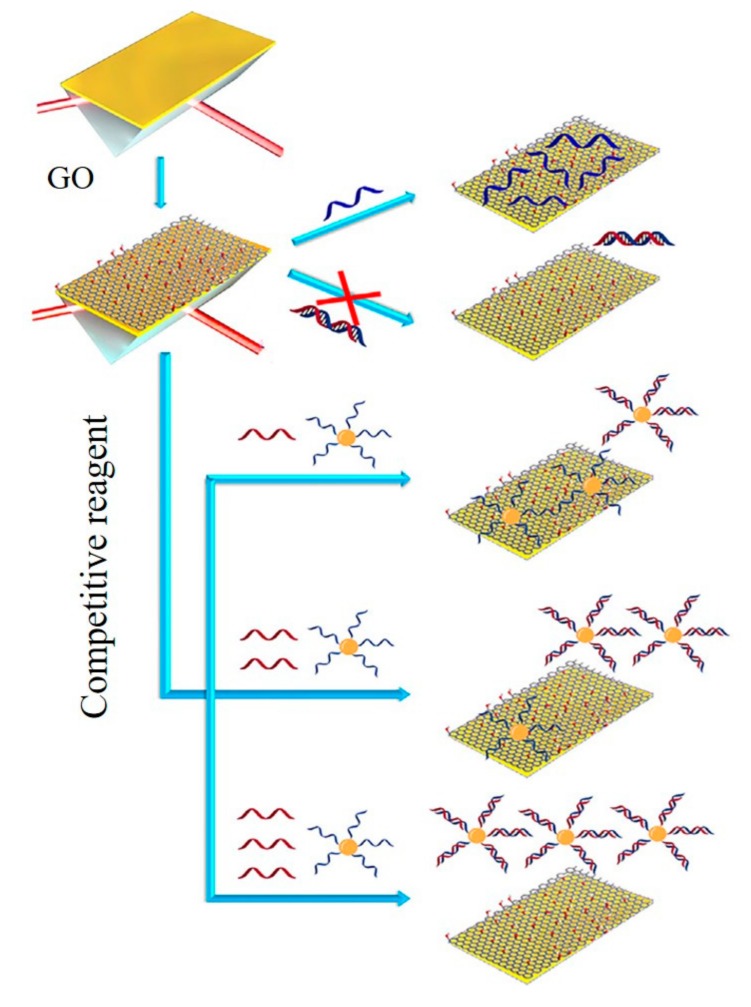
GO-based SPR biosensor for detection of ssDNA. Reprinted with permission [69]. Copyright 2014, Elsevier B.V.

**Figure 12 micromachines-11-00060-f012:**
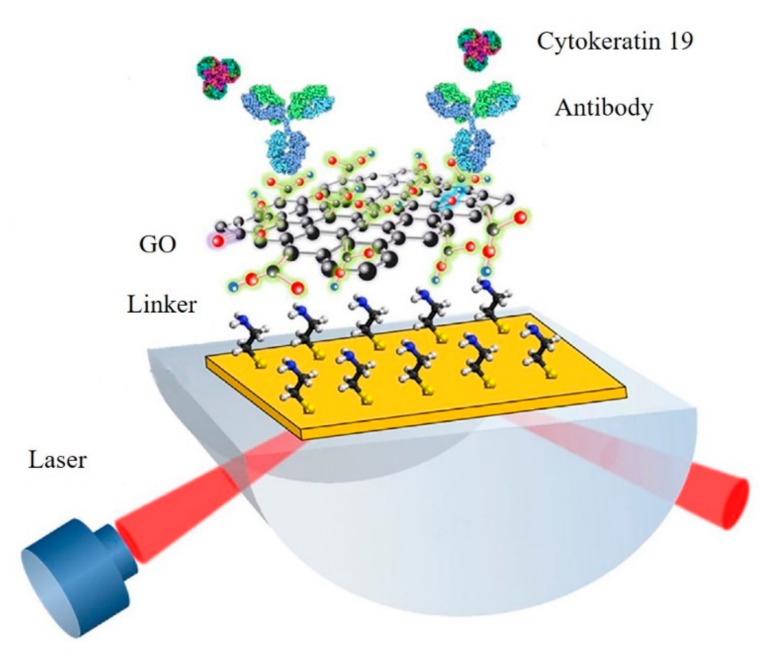
GO-based SPR immunosensor for detection of cytokeratin 19. Reprinted with permission [40]. Copyright 2018, Elsevier B.V.

**Figure 13 micromachines-11-00060-f013:**
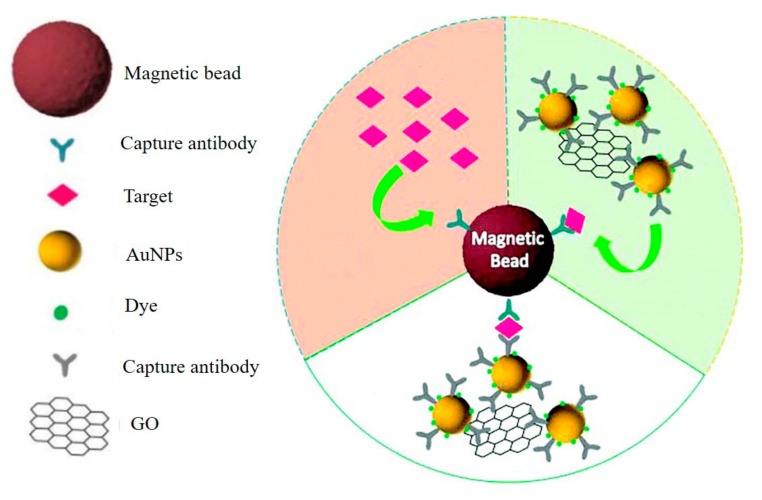
GO-AuNPs-based amplification SERS immunosensor for detection of cTnI. Reprinted with permission [44]. Copyright 2019, Royal Society of Chemistry.

**Figure 14 micromachines-11-00060-f014:**
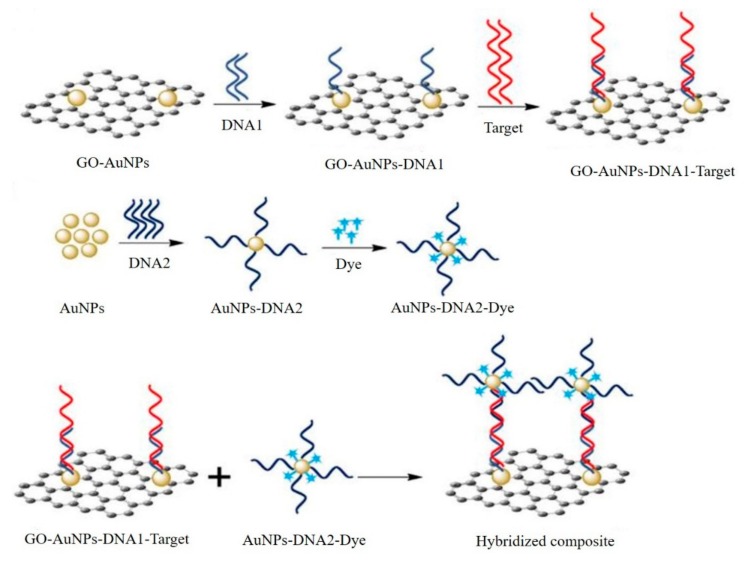
GO and AuNPs-based SERS platform for DNA biosensing. Reprinted with permission [45]. Copyright 2019, Elsevier B.V.

**Figure 15 micromachines-11-00060-f015:**
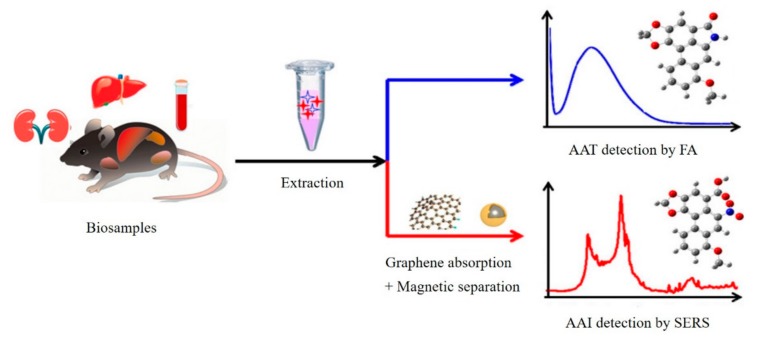
Scheme of dual-spectroscopic system for detection of aristolochic acids. Reprinted with permission [46]. Copyright 2019, American Chemical Society.

**Figure 16 micromachines-11-00060-f016:**
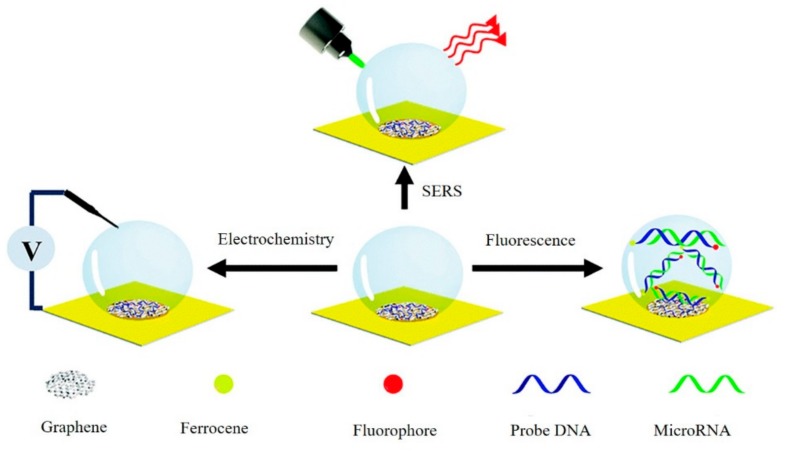
Scheme of nanodendritic gold/graphene-based biosensor for tri-mode miRNA sensing. Reprinted with permission [74]. Copyright 2019, Royal Society of Chemistry.

**Table 1 micromachines-11-00060-t001:** Graphene-based biosensors for detection of various targets.

Sensor Type	Design Method	Target	Detection Limit	Dynamic Range	Ref
Fluorescence	GO-aptamer, enzyme	ATP	31 nM	50–500 nM	[23]
Electrochemistry	Graphene-poly (3-aminobenzoic acid)	PSA	0.13 pg	0.01–80 ng/mL	[24]
Electrochemistry	rGO-Au	Folic acid	1 pM	1–200 pM	[25]
Electrochemistry	rGO (Mg/Zn)	Folic acid	25 nM	0.902–8.52 μM	[26]
Electrochemistry	*N*-graphene-polyaniline AgNPs-ssDNA	MicroRNA	0.2 fM	10 fM–10 µM	[27]
Electrochemistry	Graphene-antibody	Cystatin C	0.03 ng/mL	0.1–1000 ng/mL	[28]
Electrochemistry	GO-Fe_3_O_4_	Glucose	0.01 μM	0.05–50 μM	[29]
Electrochemistry	rGO	PSA	2 pg/mL 0.06 ng/mL	1–36 ng/mL 0.0018–41.15 ng/mL	[30]
Electrochemistry	N, S-rGO-antibody-AuNPs- AgNPs	cTnI	33 fg/mL	100 fg/mL–250 ng/mL	[31]
Electrochemistry	GO-carbon nanotubes-aptamer-hemin-	CEA	0.82 fg/mL	1 fg/mL–10 μg/mL	[32]
Electrochemistry	GO-ssDNA-poly-l-lactide	VEGF PSA	50 ng/mL 1 ng/mL	0.05–100 ng/mL 1–100 ng/mL	[33]
Electrochemistry	GO-PdNPs	Dopamine	23 nM	0.3–196.3 μM	[34]
SPR	GO-bacterial	Lysozyme	0.05 mg/mL	0.2–40 μg/mL	[35]
SPR	GO-AuNPs-ss DNAGO-AuNPs-aptamer	MicroRNA Adenosine	0.1 fM 0.1 pM	- 0.1 pM–2 nM	[36]
SPR	silver-GO silica fiber	IgG	0.04 μg/mL	-	[37]
SPR	Graphene	Folic acid	5 fM	5–500 fM	[38]
SPR	Nickel-graphene	3-nitro-l-tyrosine	0.13 pg/mL	0.5 pg/mL–1 ng/mL	[39]
SPR	GO-antibody	Cytokeratin 19	1 fg/mL	1 fg/mL–1 ng/mL	[40]
SERS	GO-AuNPs-antibody-magnetic bead	IgG	31 fM	0.1–10,000 pM	[41]
SERS	GO-AgNPs-antibody	PSA	0.23 pg/mL	0.5–500 pg/mL	[42]
SERS	GO-Aptamer-Au	ATP	0.85 pM	10 pM–10 nM	[43]
SERS	GO-Au-antibody-malachite green isothiocyanate	cTnI	5 pg/mL	0.01–1000 ng/mL	[44]
SERS	GO-AgNPs-ssDNA-cy3	ssDNA	10 fM	10 fM–10 μM	[45]
SERS	Graphene-Fe_3_O_4_-AgNPs	Aristolochic acids	100 ppb	200 ppb–10 ppm	[46]

ATP: Adenosine triphosphate, PSA: Prostate specific antigen, CEA: Carcinoembryonic antigen, VEGF: Vascular endothelial growth factor, cTnI: Cardiac troponin I, IgG: Immunoglobulin G, RNA: Ribonucleic acid, DNA: Deoxyribonucleic acid, SPR: Surface plasmon resonance, SERS: Surface enhanced Raman scattering.

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
