# Peer review of "Graphene-Based Biosensors for Detection of Biomarkers"

_micromachines, 2020, doi:10.3390/mi11010060_

Round 1
Reviewer 1 Report
Journal: Micromachines
Manuscript ID: micromachines-671963
Title: Graphene-based Biosensors for Detection of Biomarkers
In this review article, the authors reported Graphene-based Biosensors for the Efficient detection of Biomarkers. The construction of the manuscript is well organized, the tables and schemes are rigorous, and the conclusion is supported. I think that this manuscript can be accepted to publish in Micromachines.
Author Response
General Comment: In this review article, the authors reported Graphene-based Biosensors for the Efficient detection of Biomarkers. The construction of the manuscript is well organized, the tables and schemes are rigorous, and the conclusion is supported. I think that this manuscript can be accepted to publish in Micromachines.
Response: We thank the reviewer #1 for recognizing the novelty and high quality of the presented work and for recommending it as for a potentially good candidate publication in Micromachines
Reviewer 2 Report
This is a well written review of graphene biosensors. They have referenced literature adequately and will be very useful for a wide range of readers.
Author Response
General Comment: This is a well written review of graphene biosensors. They have referenced literature adequately and will be very useful for a wide range of readers.
Response: We thank the reviewer #2 for recognizing the innovative aspects in our presented work.
Reviewer 3 Report
The manuscript entitled "Graphene-based Biosensors for Detection of Biomarkers" reported a review article for the graphene-based biosensors. However, the experimental details are needed to clarify. In my opinion, the work is moderate and it can be considered after addressing the comment carefully.
Here are the comments for the authors:
It is suggested to name the atoms in Fig. 1. The subheading of 3.2 is missing. It will be interesting to include the publication number within 5 years based on 5 types (Fluorescence, FA, Electrochemistry SPR, SERS) of graphene-based sensors. In table 1, it is suggested to use the literature published within 3 years. This will be helpful to summarize the recent progress.
Author Response
General Comment: The manuscript entitled "Graphene-based Biosensors for Detection of Biomarkers" reported a review article for the graphene-based biosensors. However, the experimental details are needed to clarify. In my opinion, the work is moderate and it can be considered after addressing the comment carefully.
Response: Thanks for your suggestion, we have revised the manuscript to address all the issues as you suggested.
Comment 1. It is suggested to name the atoms in Fig. 1.
Thanks for your comments, we have fully revised our manuscript according to your suggestions. We named the atoms in Fig. 1 in Page 2.
Comment 2.The subheading of 3.2 is missing.
Response: Thanks for your comments, the subheading of 3.2 has been corrected.
Comment 3. It will be interesting to include the publication number within 5 years based on 5 types (Fluorescence, FA, Electrochemistry SPR, SERS) of graphene-based sensors. In table 1, it is suggested to use the literature published within 3 years. This will be helpful to summarize the recent progress.
Response: The publication number within 5 years based on 5 types (Fluorescence, FA, Electrochemistry SPR, SERS) of graphene-based sensors is about 10000 publications. However, there are only 230 publications about biosensor (date from web of science). We addressed such point in the Section 3 (Page 3). We also revised the table 1 as you suggested in Page 3.
Round 2
Reviewer 3 Report
The authors have taken the reviewers' comments seriously and revised the
manuscript accordingly. In my opinion, the revised manuscript could be
considered for publication in the journal without further modification.